# [Reproducibility Report] Normalized Loss Functions for Deep Learning with Noisy Labels

## Reproducibility Summary

*The central claim of paper is that Normalized Loss functions called "Active Passive Loss" perform better on datasets with noisy labels. We have tried to reproduce the result on one of the metric on CIFAR-10 dataset. On training on 120 epochs and same hyper parameters mentioned on the paper, the result obtained has similar testing accuracy claimed in the paper.*

## Scope of Reproducibility

Scope of reproducibility is to train the CIFAR-10 dataset with noisy labels of 4% for the metric NCE+RCE and check the claim of the test accuracy of the metric NCE+RCE mentioned in the paper.

## 1 Methodology

We have used author's code. It took around 23 hours to train the model for 120 epochs.

### 1.1 Datasets

Data used was CIFAR-10. We obtained the dataset by modifying the author's code to download it onto the local drive.

### 1.2 Hyperparameters

We have used the hyperparameters mentioned in their code like

- Epoch:120
- Optimizer : SGD optimizer with momentum 0.9 and weight decay 0.0001
- alpha,beta : 1.0

### 1.3 Experimental setup

We were trying to execute the code and reproduce the results on different hardware/software environments like different environments like Local CPU, Google Colab Notebook and 2 kind of AWS GPU systems with different setup. The code failed to run over certain epochs in CPU machine (RAM: 16 GB)owing to the GPU requirement of the training procedure. We were not able to complete the experiment on a Google Colab because the runtime exceeded 12 hours limit. We successfully run the experiment of 120 epochs on AWS Deeplearning AMI machine (CPU RAM : 32 gb, 8 vCPU and 2 GPU - 8gb ).

### Results

we reproduced the accuracy to within 1% of reported value(ours - 85.97% , paper's conclusion - 86.02%) for same number of epochs , with noisy labels of 4% on same data set and code, that upholds the paper's conclusion and claim that NCE+RCE performs much better than baselines

## What was easy

It was easy to run the author's code. It was well structured and easy to run in new software/hardware setup. We were able to easily execute the code on different environments like Local CPU, Google Colab Notebook and 2 kind of AWS GPU systems with different setup.

## What was difficult

We had to make small changes to the code to download the required data set. We spent some time on checking the necessary compute resources required by trying different machines.

## Communication with original authors

We did not communicate with the original authors.

