# OpenReview forum: "[Re] Normalized Loss Functions for Deep Learning with Noisy Labels"
_ML_Reproducibility_Challenge/2020 — Reject_

### Official Review · AnonReviewer2 · 2021-02-20
**Extremely limited scope, does not provide additional information over the original article**

**Rating:** 3
**Confidence:** 4

**Review:**

Limited scope
--------------------
The authors clearly describe the scope and adhere to it, however, the scope is extremely limited:
- one dataset only (CIFAR-10), when the original article used 2 other small-scale ones (MNIST and CIFAR-100) and a larger-scale one (mini-ILSVRC12)
- one loss combination (NCE+RCE), when the original article tested 3 other ones extensively as well as 5 baselines, and has additional results for 2 other variants
- one noise setting (symmetric with a noise rate of 0.4) when the original article reports most results on 9 (1 clean, 4 symmetric, 4 asymmetric)

The authors also did not justify why they chose that specific combination, out of the 48 bolded ones in original tables 2 and 3 (among 972 entries when including baselines and "clean" experiments).

The only source of variation envisioned in the experimental setup was running the code on different hardware and environments: local CPU, Google Colab, and AWS GPU, all using the same software.

Limited effort
-------------------
Of the different hardware settings, only one result was reported, and there was no detailed analysis of failures, or mention of mitigation strategies. For instance:
- "the code failed to run over certain epochs in CPU machine": in what way? Could it have worked on a smaller dataset (e.g., MNIST), or maybe with a smaller batch size?
- the runtime exceeded the 12 hours limit on Colab: maybe the authors could have implemented a checkpointing strategy, and restarted training a few time.

In addition, the report contains:
- no summarization or explanation of the original paper, methods, or experimental setting
- no ablation studies: only one model was retrained as described, the baselines were not retrained
- no hyperparameter search: only the values found in the code were used
- one single run with a given random seed, even though the original article reported mean±std over 3 runs for most results

Confusing reporting of results
------------------------------------------
The symmetric noise setting with a parameter of $\eta = 0.4$ is characterized as "noisy labels of 4%" when the original article indicate it would be 40% of corruption ($1 - \eta$ or 60% of labels would stay the same).

The original result is reported as "86.02%", omitting the standard deviation of "± 0.09".
The reproduced result of 85.97% is reported as "within 1%" of the original one, which is technically correct as $0.05 < 1$, but confusing. Why mention a 1% threshold? Was it determined in advance?

Conclusion
----------------
In the end, this report does not shed any additional light on the original paper or code base, except that the code runs and did not produce an unexpected result.

The [task description](https://paperswithcode.com/rc2020/task) for the challenge mentions: _"Just re-running code is not a reproducibility study"_, and I don't think this reports clears the bar, I recommend to reject it.



**Familiar With The Original Paper:**

I have read the original paper

**Reproducibility Summary:**

Report has summary

---

### Official Review · AnonReviewer3 · 2021-03-09
**Reproducibility**

**Rating:** 5
**Confidence:** 3

**Review:**

Issues regarding the report:
Authors test the code only on the CIFAR-10 with 0.4 Symmetric Noise Rate. I believe it would be better to show at least one more experiment (maybe with 0.8).

Authors need to give a better explanation of how hyperparameters are selected.  Further hyperparameter search can be useful, authors can change model optimizer and learning.

Another important point, there is no information regarding the stopping criteria?. It will be better to plots how training loss changes during training.  It can be also better how to test accuracy change during training,

If possible authors can train and test one of the competing models as well. We can see how improvements happen.  Maybe improvement related to the optimizer or something else?


**Familiar With The Original Paper:**

I have read the original paper

**Reproducibility Summary:**

Report has summary

---

### Decision · Program_Chairs · 2021-03-31

**Decision:**

Reject

**Comment:**

Overall reviews and/or the paper content not good enough for the AC to recommend to the journal.